# Synthesis and Properties of α-Phosphate-Modified Nucleoside Triphosphates

**DOI:** 10.3390/molecules29174121

**Published:** 2024-08-30

**Authors:** Alina I. Novgorodtseva, Alexander A. Lomzov, Svetlana V. Vasilyeva

**Affiliations:** Institute of Chemical Biology and Fundamental Medicine, SB RAS, 8 Lavrentiev Avenue, Novosibirsk 630090, Russia; novgorodtseva@niboch.nsc.ru

**Keywords:** α-phosphate mimetic, α-P-modified nucleoside triphosphate, cyclotriphosphite intermediate, 5′-(α-P-thio)nucleoside triphosphate, 5′-(α-P-seleno)nucleoside triphosphate, 5′-(α-P-borano)nucleoside triphosphate, 5′-(α-P-alkyl)nucleoside triphosphate, 5′-(α-P-amido)nucleoside triphosphate, 5′-(α-P-N-alkyl)nucleoside triphosphate, 5′-(α-P-imido)nucleoside triphosphate, nucleic acid metabolism enzyme, nuclease resistance

## Abstract

This review article is focused on the progress made in the synthesis of 5′-α-P-modified nucleoside triphosphates (α-phosphate mimetics). A variety of α-P-modified nucleoside triphosphates (NTPαXYs, Y = O, S; X = S, Se, BH_3_, alkyl, amine, N-alkyl, imido, or others) have been developed. There is a unique class of nucleoside triphosphate analogs with different properties. The main chemical approaches to the synthesis of NTPαXYs are analyzed and systematized here. Using the data presented here on the diversity of NTPαXYs and their synthesis protocols, it is possible to select an appropriate method for obtaining a desired α-phosphate mimetic. Triphosphates’ substrate properties toward nucleic acid metabolism enzymes are highlighted too. We reviewed some of the most prominent applications of NTPαXYs including the use of modified dNTPs in studies on mechanisms of action of polymerases or in systematic evolution of ligands by exponential enrichment (SELEX). The presence of heteroatoms such as sulfur, selenium, or boron in α-phosphate makes modified triphosphates nuclease resistant. The most distinctive feature of NTPαXYs is that they can be recognized by polymerases. As a result, S-, Se-, or BH_3_-modified phosphate residues can be incorporated into DNA or RNA. This property has made NTPαXYs a multifunctional tool in molecular biology. This review will be of interest to synthetic chemists, biochemists, biotechnologists, or biologists engaged in basic or applied research.

## 1. Introduction

Enzymatic polymerization of modified triphosphate represents a versatile platform for easy preparation of functionalized nucleic acids. Nucleoside triphosphates are building blocks that can be readily functionalized at a variety of sites [1]. Natural nucleic acids lack diversity of functional groups and have low resistance to nucleases. These drawbacks have prompted the development of ligand-finding systematic evolution of ligands by exponential enrichment (SELEX) and related in vitro selection methods involving modified triphosphates [2,3]. In cellular processes, NTPs can serve as intermediates in the biosynthesis of nucleosides, proteins, and hormones [4]. Current applications of NTP analogs include DNA sequencing [5], the creation of therapeutic nucleotide-based inhibitors of polymerases [6], the production of aptamers [7], polymerase chain reaction (PCR) diagnostics [8], research into mechanisms of enzymatic reactions [9], and other uses. Therefore, systematic information about methods for the synthesis (and properties) of modified triphosphates is important and relevant.

As a key class of nucleoside triphosphate analogs, α-P-modified nucleoside triphosphates (NTPαXYs) have long been recognized as versatile tools in molecular biology and medicinal chemistry [10]. The α-phosphate moieties are especially important because after cleavage of the pyrophosphate, phosphate-modified units are incorporated into a long nucleic acid strand (Figure 1). In comparison to β- and γ-P-modified NTPs, the most distinct feature of NTPαXYs is their recognition by polymerases. This feature makes it possible to incorporate S-, Se-, or BH_3_-modified phosphate linkages into DNA and RNA. As a result, NTPαXYs have become a multifunctional tool for various purposes. For instance, it is known that nucleoside 5′-(*α*-P-thio)triphosphates (NTP*α*-Ss) [11] and nucleoside 5′-(*α*-P-borano)triphosphates (NTP*α*-Bs) [12] can be easily incorporated into nucleic acids using DNA or RNA polymerases [13].

The presence of heteroatoms, such as sulfur, selenium, and boron, in the α-phosphate moiety endows these triphosphates with excellent resistance to nucleases [10].

The resulting chirality at the α-P position has enabled these molecules to function as useful probes in the investigation into mechanisms of action of NTP-binding enzymes [12,14,15]. DNA and RNA polymerases “recognize” only certain diastereomers of nucleotide analogs, for example, Sp-diastereomers of dNTPαS and NTPαS, as demonstrated in the work of J. Caton-Williams et al. [16].

Compared to the synthesis of regular NTPs [1,17,18], the synthetic methods for NTPαXYs are rather limited due to the introduction of heteroatoms at the α-P position.

The present review provides information on chemical methods for the synthesis of nucleoside triphosphates modified on α-phosphate (α-phosphate mimetics). The best-known and most widely used of them are 5′-(α-P-thio)-, 5′-(α-P-seleno)-, and 5′-(α-P-borano)triphosphates. There are much fewer data in the literature on techniques for the synthesis of 5′-(α-P-alkyl)triphosphates and 5′-(α-P-amido)triphosphates. Very recently, a method was devised for the preparation of 5′-(α-P-imido)triphosphates via Staudinger’s reaction. Methods are also presented here regarding the synthesis of triphosphates simultaneously modified on 5′-α-phosphate and the glycosidic residue or heterocyclic base as well as featuring replacement of two oxygens of 5′-(α-P)phosphate with different or identical groups simultaneously. For the 5′-α-P-modified nucleoside triphosphates in question and oligonucleotides derived from them, their substrate properties are discussed in terms of nucleic acid metabolism enzymes.

## 2. Chemical Synthesis of Nucleoside 5′-α-P-Modified Triphosphates

### 2.1. Synthesis of Nucleoside 5′-(α-P-Thio)- and 5′-(α-P-Seleno)triphosphates

In this subsection, along with approaches to the synthesis of 5′-(α-P-thio)triphosphates, data on 5′-(α-P-seleno)triphosphates are presented too because the same methods have often been used for their preparation, with the only difference being which reagent is applied at the oxidation stage: a sulfurizing or selenizing one.

#### 2.1.1. Syntheses via a Nucleophilic Attack of Pyrophosphate on Activated Nucleoside 5′-Phosphorothioates

Nucleoside 5′-(α-P-thio)triphosphates were initially obtained by activation of nucleoside 5′-phosphorothioates followed by the addition of pyrophosphate. For the first time, the preparation of 5′-(α-P-thio)nucleoside triphosphate by a chemical method was proposed by F. Eckstein in 1967 [19]. Thymidine 5′-(α-P-thio)triphosphate **1** was synthesized with 5% yield by the so-called triimidazolyl-phosphine sulfide method, which involved a reaction of 3′-*O*-acetyl-thymidine with an excess of triimidazolyl-1-phosphine sulfide. The product was thymidine 5′-phosphoromonoimidazolothioate, which was then treated with pyrophosphate to obtain desired product **1** (Figure 2).

In the synthesis of adenosine 5′-(α-P-thio)triphosphate **2** (Figure 3), diphenyl phosphorochloridate has been used to activate adenosine-5′-phosphorothioate, thereby allowing to increase the yield to 20% [20]. Nonetheless, the procedure involves isolation of an intermediate product and takes at least 5 h.

Nucleoside 5′-(α-P-thio)triphosphates can be obtained with up to 25% yield directly from nucleosides via treatment with thiophosphoryl chloride and subsequent addition of pyrophosphate without isolation of an intermediate [21] (Figure 4). This technique is a version of Yoshikawa’s method [22], in which thiophosphoryl chloride is used instead of phosphorus oxychloride.

The above method is especially suitable for the synthesis of radio-labeled triphosphate analogs because chemical manipulations are minimal. The technique [22] can also be modified to introduce oxygen isotopes into the α-phosphate group. This is easy if after mixing with pyrophosphate, the reaction is stopped by the addition of isotope-labeled water. The same approach was used in the work of M. Auer et al. [23] for the synthesis of 5′-(α-P-thio)triphosphates of dideoxynucleosides, with the total yield of 15–20%.

The authors of the work [24] (Y. Thillier et al.) were the first to describe a convenient and effective technique for obtaining 5′-(α-P-thio)triphosphates of oligonucleotides on a solid support. Their strategy is similar to the one described above [20]: activation of 5′-phosphorothioate with diphenyl phosphorochloridate, with subsequent substitution by pyrophosphate (Figure 5). The 5′-(α-P-Thio)triphosphate and 5′-(α-P-thio)(β,γ-methylene)triphosphate oligonucleotides **3** were prepared in good yields ranging from 30% to 50%. Of note, this method proved to be more successful for obtaining 5′-(α-P-thio)triphosphate oligonucleotides on a solid phase as compared to other techniques, involving the use of phosphoramidate or salicylphosphite intermediates [24].

In 2005, K. Misiura et al. proposed the oxathiaphospholane approach, in which 5′-(α-thio)triphosphates of various ribo- and deoxynucleosides (A, C, G, or T/U) **6** were prepared by reacting corresponding protected nucleoside 5′-(2-thio-1,3,2-oxathiaphospholanes) **4** with pyrophosphate in the presence of diazabicycloundec-7-ene (Figure 6) [25]. The protected nucleoside reacts with 2-chloro-1,3,2-oxathiophospholane in the presence of elemental sulfur resulting in oxathiaphospholane derivative **4** in high yields. Oxathiaphospholanes (a mixture of diastereomers, approximately 1:1) are in turn reacted with tributylammonium pyrophosphate, resulting in protected nucleoside 5′-(α-P-thio)triphosphates **5** with 58–78% yields for different nucleobases. The presented method also allows to implement α-selenium modifications and to obtain corresponding α-modified diphosphates [25].

A similar method based on 2-chloro-1,3,2-dithiaphospholane or 2-(N,N-diisopropylamino)-1,3,2-dithiaphospholane and protected deoxynucleosides (cytidine, guanosine, or adenosine) as well as 3′-azidothymidine enabled us to prepare the corresponding 5′-(α-P-dithio)triphosphates [26].

#### 2.1.2. Synthesis of Nucleoside Triphosphates via a Cyclotriphosphite Intermediate (Ludwig–Eckstein Method)

Since 1989, the approach proposed by F. Eckstein and J. Ludwig has been used predominantly for the synthesis of 5′-(α-P-thio)triphosphates [27]. The key to this approach is the formation of 5′-cyclotriphosphite **9**. Salicyl chlorophosphite has been utilized as a phosphitylation reagent (Figure 7), and the reaction with nucleoside **7** gives two diastereomers **8**. Next, after the addition of pyrophosphate, the formation of cyclic intermediate **9** takes place. This intermediate in the presence of elemental sulfur gives rise to 5′-α-P-thio-cyclotriphosphite **10**, aqueous hydrolysis of which produces 5′-(α-P-thio)triphosphate **11**. The authors confirmed that the ring opening occurs precisely at the bond between α-phosphate and γ-phosphate through hydrolysis with H_2_^18^O. Considering this, cyclic intermediate **9** can be oxidized with an aqueous solution of iodine to obtain native triphosphates **12**.

The advantage of this technique is that protection of functional groups of nitrogenous bases is not required. This method is applicable to the synthesis of eight common ribo- and deoxyribonucleotides, with an overall purification yield of 60–75%.

Caton-Williams et al. [28] reported a method for the synthesis of triphosphates with α-P-thio and α-P-selenium substitutions **15** and **16** (Figure 8). Sulfurization of nucleoside cyclic phosphite **14** is performed using 3-[(dimethylaminomethylidene)amino]-3H-1,2,4,dithiazole-3-thione (sulfurizing reagent). Subsequent hydrolysis affords crude dNTPαS and NTPαS analogs **15** as mixtures of Sp- and Rp-diastereomers with yields up to 60%. When this synthetic strategy is applied, only one diastereomer of dGTPαS and GTPαS is observed. Nucleoside 5′-(α-P-seleno)triphosphates (dNTPαSe and NTPαSe) **16** are prepared by replacing the iodine solution at the oxidation step with selenium-introducing oxidant 3H-1,2-benzothiaselenol-3-one. The 5′-triphosphate products are diastereomeric mixtures, and the synthesis yields are generally greater than 30%. Protection of nitrogenous bases and hydroxyl groups of the ribose residue is not required owing to the use of reagent **13** obtained beforehand (Figure 8) because reagent **13** selectively phosphorylates the 5′-OH group of the nucleoside. Later, this method in the “one-pot” format without protection of the functional groups of ribonucleosides is employed to synthesize analogs of 5′-(α-P-seleno)nucleoside triphosphates (NTPαSs) [29] (32–67% yields achieved) and 5′-(α-P-thio)nucleoside triphosphates (NTPαSs) [16] (yields of 45–60% attained).

Thymidine 5′-(α-P-borano, α-P-thio)triphosphate **21**, in which borane and sulfur simultaneously replace the two nonbridging oxygens of α-phosphate, was also obtained by the salicyl chlorophosphite method (Figure 9) [30]. The introduction of the borane group into cyclophosphate **18** was performed with an excess of the borane–diisopropylamine complex. After that, cyclophosphate with a borano group (**19**) is treated with lithium sulfide, which introduces a thio group and opens the ring, resulting in 5′-triphosphate **20**. The overall yield of thymidine 5′-[a-P-borano, a-P-thio]triphosphate **21** is approximately 26%. The presented synthetic approach can be applied to the preparation of any (α-P-borano or α-P-thio)-modified NTP or dNTP.

In the work of A. Ripp et al. [31], instead of salicyl chlorophosphite, pre-prepared cyclic pyrophosphoryl-P-amidite **22** was used as the phosphitylation reagent (Figure 10). Under dry conditions and in the presence of an activator, it reacts with 3′-azidothymidine **23**, again generating cyclotriphosphite intermediate **24**. Next, there are three possible options for oxidation of compound **24**: {1} oxidation with meta-chloroperbenzoic acid to obtain **25a**; {2} oxidation using Beaucage’s reagent to obtain thiocyclotriphosphate **25b**; or {3} use of potassium selenocyanate (KSeCN) to synthesize respective selenium derivative **25c**. The resulting cycloesters **25a**–**c** are then opened by means of a nucleophile. When propargylamine serves as the nucleophilic reagent, 3′-azidothymidine 5′-γ-P-propargylamidotriphosphate (α-P-S or α-P-Se) derivatives **26a**–**c** are generated. In most cases, the product is obtained with high purity; precipitation in the form of sodium salts is enough for purification. Compound **25b** is obtained as a mixture of diastereoisomers in a ∼1:1 ratio with 59% yield; compound **25c** is obtained with 40% yield.

#### 2.1.3. Synthesis by the Amidophosphite Method

Research of Qi Sun et al. [10] revealed that linear P^V^P^V^P^III^-nucleoside intermediates can be efficiently generated via the coupling of a specific type of nucleoside 5′-phosphoramidites (fluorenylmethyl nucleosidylphosphoromorpholidites) with pyrophosphate in the absence of activators. In situ fast oxidation with chalcogens and borane dimethyl sulfide followed by Fm cleavage and nucleoside deprotection provided a novel and efficient route to NTPαXs (X = S, Se, or BH_3_) **27a**–**c** (Figure 11). NTPαSs were prepared with 66–76% yields. NTPαSs were prepared with 54–63% yields, and NTPαBs with 67–70% yields. Furthermore, this new method has been successfully extended to the synthesis of α-P-modified dinucleoside tetraphosphates.

### 2.2. Synthesis of Nucleoside 5′-(α-P-Borano)triphosphates

#### 2.2.1. Synthesis by the Amidophosphite Method

In the early 1990s, Sood’s laboratory for the first time prepared thymidine 5′-(α-P-borano)triphosphate (TTPαB) **30** using boranoamidophosphate **28** as a key intermediate (Figure 12) [32]. The synthesis was carried out through boration of thymidine amidophosphite **27** with the formation of complex **28**, which was then converted to boranoamidophosphate **29** via β-elimination of the cyanoethyl group. Compound **29** reacted with excess tributylammonium pyrophosphate salt [(Bu_3_NH)_2_H_2_P_2_O_7_], giving a mixture of diastereomers of TTPαB **30** with 30% yield after purification by ion exchange chromatography. This technique can also be used to prepare other dNTPαB, but protection of the exocyclic amino group in nitrogenous bases and of the 3′-hydroxyl group is required.

#### 2.2.2. Synthesis through a Cyclotriphosphite Intermediate

Another version of the synthesis of 5′-(α-P-borano)triphosphate of adenosine and its analogs was developed by V. Nahum et al. [33] (Figure 13). For phosphitylation of the protected nucleoside, bis-diisopropylaminochlorophosphine was used, and therefore this method may be classified as an “amidophosphite” technique. Nonetheless, 5′-cyclotriphosphite **31** was next obtained from it, which was then borated to replace oxygen with various borane complexes. Borating with a 2 M solution of borane dimethyl sulfide (Me_2_S:BH_3_) in tetrahydrofuran (THF) gave the best results, with an overall yield of 31–43% of compound **32**. Advantages of this method are as follows: there is no need to protect the purine nitrogenous base, during the synthesis there is no need to purify intermediates, and the synthesis conditions are applicable to adenosine analogs with a modification in the heterocycle (Figure 13).

A similar technique was employed to obtain a 5′-(α-P-borano)triphosphate derivative of acyclothymidine **33** [34]. The method involves tetrazole as a catalyst for the reaction of substitution of diisopropylamide groups with pyrophosphate (Figure 14). After final purification, the yield of product **33** was 53%.

Shaw’s laboratory devised a convenient one-pot method for synthesizing a series of ribonucleoside 5′-(α-P-borano)triphosphates (A, U, G, and C) (Figure 15) [35]. Phosphitylation of a 2′,3′-protected ribonucleoside with salicyl chlorophosphite and a pyrophosphate exchange reaction give an intermediate product: cyclotriphosphite, just as in the previously designed method [27]. To obtain nucleoside boranocyclotriphosphate **34**, Me_2_S:BH_3_ was used as a borating reagent. Next, the cycle was opened with water, and subsequent hydrolysis with ammonia was conducted. Introduction of a borane group (BH_3_) to replace one of the oxygen atoms on the α-phosphate produced a pair of NTPαB diastereomers **35**. These two diastereomers were separated by preparative reverse-phase high-performance liquid chromatography (HPLC). The ratio of isomers varies slightly depending on the nucleotide, for example, for UTPαB, it is 46.4:53.6; for ATPαB, 42.7:57.3; for GTPαB, 47.5:52.5; and for CTPαB, it is 51.0:49.0. The overall yield of **35** after purification by ion exchange chromatography is 30–45%.

This technique is suitable for the preparation of 5′-(α-P-borano)triphosphates of ribo- and deoxyribonucleosides [36]. A series of NTPαB 5-substituted analogs of 2′-deoxycytidine (such as 5-methyl, ethyl, bromine, and iodine) was obtained in the work of B.R. Shaw et al. [37] via the same approach.

In a similar way, Wang et al. synthesized 3′-azidothymidine-5′-triphosphate mimetics with 5′-α-P-borano, 5′-α-P-thio, or 5′-α-P-dithio substitutions **36a**–**c** with moderate-to-good yields [38] (Figure 16), and in work [6], also with 5′-α-P-selenium substitutions.

### 2.3. Synthesis of Nucleoside 5′-(α-P-Alkyl)triphosphates

To obtain nucleoside 5′-(α-P-alkyl)triphosphates, activation of nucleoside 5′-O-(alkyl)phosphonates and subsequent addition of pyrophosphate are performed. Dineva et al. used diphenyl phosphorochloridate to activate thymidine 5′-methylphosphonate (Figure 17) [39]. The reaction took place within 15 min. The activated intermediate product was isolated and then treated with pyrophosphate. The yield of thymidine 5′-(α-P-methyl)triphosphate **37** was 28%.

The method of activation of thymidine 5′-P-methylphosphate by means of carbonyldiimidazole was utilized by the authors of one study [40], where they obtained thymidine 5′-*O*-(methylphosphoimidazolide) and then—through condensation with pyrophosphate—thymidine 5′-(α-P-methyl)triphosphate with 20–30% yield.

This technique is suitable for the synthesis of thymidine 5′-(α-P-methyl)triphosphate but cannot be applied to similar compounds with other nitrogenous bases (cytidine, adenosine, and guanosine); their synthesis requires protection of the exocyclic amino group, but standard deprotection conditions are incompatible with the methyl phosphate modification. One solution to this problem is chemoenzymatic synthesis based on phenylacetyl protection of exocyclic amino groups, which is cleaved off by an enzyme [41] (Figure 18). The protecting group is easily removed by treatment with penicillin amidase at pH 7.8 and 25 °C. The reaction with phenylacetyl chloride is implemented through protection of 3′- and 5′-OH groups beforehand with trimethylchlorosilane in the presence of benzotriazole phenylacetate as a mild acylating reagent (Figure 18).

A reaction of a phenylacylated derivative of a nucleoside with methylphosphonic acid and 1,3-dicyclohexylcarbodiimide (DCC) in pyridine gives a 5′-P-methylphosphate derivative of the nucleotide. Pyrophosphorylation in the presence of carbonyldiimidazole as a condensing agent allows to obtain 5′-(α-P-methyl)triphosphates of cytidine, guanosine, and adenosine **38a**–**c** in yields of 80–85%.

When deprotection is performed with penicillin amidase, the α-P-methylphosphate group and phosphoanhydride bonds remain stable even after a long period of deprotection. As a consequence, 5′-(α-P-methyl)triphosphates of deoxynucleosides are formed, which can be used without further purification as substrates for PCR.

A. Arzumanov and N. Dyatkina [42] proposed alternative one-pot synthesis of 5′-(α-P-methyl)triphosphate of thymidine and of 3′-azidothymidine **41a**–**c** (Figure 19). The phosphorylation of the nucleoside was performed with methylphosphonium dichloride. This reaction proceeds for 10–15 h, and then condensation with pyrophosphate gives the corresponding 5′-(α-P-methyl)thymidine triphosphate **41a,c** with 28–30% yield. Replacing the phosphorylating reagent with bis-1,2,4-triazolyl methylphosphonium made it possible to shorten the reaction duration by 10-fold and to increase the yield of product **41b,c** to 32%.

Later, by a similar method involving dichloroethylphosphonium, Holliger’s research group obtained 5′-(α-P-ethyl)triphosphates of 2′-deoxyadenosine, deoxycytidine, deoxyguanosine, and thymidine [43], unfortunately, with very low yields: 1.0–1.5%.

### 2.4. Synthesis of 5′-(α-P-Amido)triphosphates and 5′-(α-P-Alkylamido)triphosphates of Nucleosides

The 5′-(α-P-Amido)triphosphates were first obtained in 1976 [44]. The gist of the method was to substitute one of the amide groups in nucleoside 5′-phosphorodiamidate with pyrophosphate (Figure 20).

A solution of thymidine 5′-phosphorodiamidate was treated with an excess of tributylammonium pyrophosphate to generate thymidine 5′-(α-P-amido)triphosphate **42b** as the main product. The yield was 72%. In the synthesis of 5′-(α-P-amido)triphosphate adenosine **42a** under similar conditions, the yield was 57%.

Those authors noticed that the resulting thymidine 5′-(α-P-amido)triphosphate was more susceptible to alkaline hydrolysis as compared to the unmodified triphosphate. This finding is explained by the fact that the uncharged α-phosphate atom has lower electron density, and pyrophosphate has properties of a good leaving group. Therefore, thymidine 5′-(α-P-amido)triphosphate can be easily attacked by nucleophilic reagents. Under slightly alkaline conditions (e.g., 1N ammonium hydroxide, 20 °C), it gets hydrolyzed to thymidine 5′-amidomonophosphate and inorganic pyrophosphate.

Attempts to synthesize ribonucleoside 5′-(α-P-amido)triphosphate and diamidophosphates have been unsuccessful [45]. Derivatives of oligoribonucleotides with amidophosphate groups are unstable due to the presence of a 2′-OH group; for example, in an aqueous solution, dinucleotide Up(NH_2_)U decomposes (with loss of NH_3_) partly into UpU and partly into U and pU. Accordingly, the 2′-OH group has to be protected; for instance, 2′-OMe nucleosides are employed for this purpose. The authors of patent [46] oxidize a nucleoside 5′-cyclotriphosphite (attached to a solid support) using a mixture of iodine with a primary amine or alkylamine; the oxidized intermediate is hydrolyzed with water, resulting in, e.g., 2′-deoxythymidine-5′-(α-P-amido)triphosphate **43a**, 2′-OMe-guanosine-5′-(α-P-N-methyl)triphosphate **43b**, or 2′-OMe-guanosine-5′-(α-P-N-butyl)triphosphate **43c** (Figure 21).

### 2.5. Synthesis of Nucleoside 5′-(α-P-Imido-R)triphosphates by Means of Organic Azides N_3_-R

Not so long ago, novel nucleoside 5′-(α-P-XY)triphosphates, namely 5′-(α-P-imido)triphosphates with substituents Y = OH and X = N-R, were prepared, where R is an electron acceptor group (Figure 1). These triphosphate analogs were obtained via Staudinger’s reaction [47]. This is a reaction of a trivalent phosphorus atom with an organic azide, preferably having an electron-withdrawing substituent (acyl azides, sulfonyl azides, or phosphoryl azides). As a result, pentavalent phosphorus arises that is covalently bonded through a nitrogen atom to a strong electron-withdrawing group. This structure, unlike phosphoramidates [44], is resonance-stabilized and much less susceptible to alkaline hydrolysis.

In 2008, as an example, the authors of patent [48] described a method, similar to the one presented in paper [27], for the synthesis of thymidine 5′-(α-P-imido)triphosphate using acetamidobenzenesulfonyl azide **46** (Figure 22). Salicyl chlorophosphite was utilized as a phosphitylation reagent. Thymidine **44** protected on the 3′-hydroxyl group was added to salicyl chlorophosphite, then the tributylammonium salt of pyrophosphate along with triethylamine was introduced, and after 1 h, the formation of cyclic intermediate **45** took place. This intermediate in the presence of 4-acetamidobenzenesulfonyl azide gave rise to 5′-α-P-imido-cyclotriphosphite, aqueous hydrolysis of which (and deprotection of the 3′-OH group) produced 2′-deoxythymidine mono-(4-acetamidobenzenesulfonyl)imido-triphosphate **46**.

This patent also describes an example of chemoenzymatic synthesis of thymidine 5′-(α-P-imido-cyan)triphosphate. First, thymidine 5′-(cyano-imido)phosphate was obtained chemically through Staudinger’s reaction with cyanogen azide, and the synthesis of the triphosphate based on it was implemented by the enzymatic method. The product, however, was not fully characterized. In particular, there are no data from ^31^P-NMR spectroscopy.

Lately, S. Vasilyeva et al. developed protocols for the synthesis of 5′-(α-P-imido)-modified mononucleotides [49]. Figure 23 shows the synthesis of 5′-(α-P-dimethylimidazolyl)phosphate of thymidine and adenosine (**47a,b**) from 5′-phosphoramidites of protected nucleosides, with 34–64% yields. The phosphoramidite was converted to 5′-triester phosphite, which interacts via Staudinger’s reaction with 2-azido-1,3-dimethylimidazole, yielding, after deprotection, nucleoside 5′-(α-P-imido)triphosphates **47a,b**.

Resulting compound **47**—owing to specific structure of the modification—is in equilibrium between imido and amido forms (Figure 24).

Thymidine 5′-(α-P-1,3-dimethylimidazolidin-2-ylidene)triphosphate (TTPαDMI) **50** (Figure 25) was synthesized through the activation of thymidine 5′-(α-P-dimethylimidazolyl)monophosphate **47a** with trifluoroacetic anhydride according to a known procedure [50] with a yield of 3%.

The resultant mixture of Sp- and Rp-diastereomers was separated by HPLC purification. The obtained compounds were fully characterized (by ^1^H- and ^31^P-NMR spectroscopy, infrared and UV spectroscopy, and electrospray mass spectrometry).

In a review [51], cyclotriphosphate and its analogs are suggested as universal intermediates for obtaining condensed phosphates. Methods are being investigated for efficient synthesis of such structures and for converting them into a wide variety of products, including 5′-α-P-modified triphosphates of nucleosides.

All the chemical synthesis methods considered above are not stereospecific and give a mixture of Sp- and Rp-diastereomers of a nucleoside triphosphate, and the reason is that α-phosphorus at this point has four nonidentical substituents and is therefore a chiral center. The authors of work [52] proposed chemoenzymatic synthesis, in which an intermediate (a nucleoside thiophosphorodichloridate) is prepared first and then converted to a triphosphate either chemically (by treatment with pyrophosphate) or via enzymatic phosphorylation. In the latter case, the reaction proceeds stereospecifically, and only the Sp-isomer is generated. Nevertheless, those authors claim that in their proposed system, chemoenzymatic synthesis with an immobilized enzymatic reagent is not as productive as chemical synthesis.

### 2.6. Summary

To summarize, to date, two main types of synthetic approaches have been described in the literature. The first one (method 1) is the coupling of pyrophosphate with activated forms of modified NMP. The second method (method 2) is the use of cyclic P^V^P^V^P^III^-nucleosides as intermediates for the synthesis of NTPαXYs. The synthetic approach via cyclic P^V^P^V^P^III^-nucleoside intermediates (in various incarnations) developed by Ludwig and Eckstein is one of the most universal and widespread for chemical synthesis of both natural nucleoside triphosphates and those modified at various sites. This technique is overall reliable and generates relatively few byproducts. It can be said that the basis for most of known methods for the synthesis of 5′-(α-P-thio, α-P-seleno-, and α-P-borano)triphosphates—starting with the approach proposed in 1989 [27]—is the preparation of a cyclophosphite intermediate. In this context, only phosphitylating and sulfurizing (or selenizing or borating) reagents are varied. Especially convenient are procedures that do not require protection of nitrogenous bases, where the method gives higher yields [overall yields of a 5′-(α-P-XY)-modified nucleoside triphosphate reach 80%] and products are purer. The formation of a cyclic triphosphite intermediate is key to substitute one of the nonbridging oxygens in α-phosphate with a thio, seleno, amino-N-alkyl, or borane group. Moreover, it is also possible to replace two nonbridging oxygens with different groups simultaneously. Through oxidation of a nucleoside 5′-cyclotriphosphite with a mixture of iodine and a primary amine or alkylamine, nucleoside 5′-(α-P-amido)triphosphates and 5′-(α-P-alkylamido)triphosphates have been obtained [46]. The oxidation of a cyclic triphosphite intermediate with azide is proposed by the authors of patent [48]. The method involving a cyclic triphosphite intermediate has not been applied only to the synthesis of nucleoside 5′-(α-P-alkyl)triphosphates; for this purpose, method 1—coupling of pyrophosphate with activated forms of modified NMP—has been employed.

Summarized data on the synthesis of various α-modified nucleoside triphosphates are presented in Table 1.

## 3. Substrate Properties of 5′-α-P-Modified Nucleoside Triphosphates and of Oligonucleotides Synthesized from Them in Relation to Nucleic Acid-Processing Enzymes; Applications of α-Phosphate-Modified Nucleoside Triphosphates

Analogs of nucleoside triphosphates of natural origin occupy a prominent place in the toolbox used by biochemists to research the functions and mechanisms of action of enzymes. These compounds can serve as analogs of substrates in the form of, e.g., reversible and irreversible inhibitors, suicide substrates, and spectroscopic probes. All these analogs differ from a normal substrate in one or more properties, which determine both their suitability for investigation into various aspects of enzyme activity and properties of the resultant products of the enzymatic reactions.

### 3.1. Nucleoside 5′-(α-P-Thio)triphosphates

Among internucleoside modifications of oligonucleotides, phosphorothioate is the most common enzymatic modification because all four nucleoside α-thiotriphosphates are good substrates for DNA and RNA polymerases [11]. The most vivid example is the synthesis of phosphorothioate-containing DNA from dNTPαS precursors [13]. In phosphorothioates, substitution of one of the nonbridging oxygen atoms with sulfur creates chirality at the phosphorus center. Only the Sp-diastereomer of nucleoside α-thiotriphosphates is a good substrate for such polymerases as the Klenow fragment (DNA polymerase) and T7 RNA polymerase [16]. P.M. Burgers and F. Eckstein [53] stated that the stereoselectivity of DNA polymerase I for the diastereomers of dATPαS differs from that for the diastereomers of dATPβS. The enzyme recognizes only the Sp-isomer of dATPαS and never the Rp-isomer as a substrate, regardless of the cation present. Those authors have concluded that coordination of the metal ion to the α-phosphate is not necessary for binding and for the reaction to proceed. That the Rp-isomer of dATPαS is not a substrate is because the binding of α-phosphorus to the enzyme is weakened or lost. Probably, a positive group (on the enzyme) that neutralizes the negative charge on the α-phosphate to make nucleophilic attack of the incoming primer 3′-hydroxy group possible can function only for the Sp-isomer (not for the Rp-isomer) of dATPαS. In the Sp-isomer, the -P-O- function points toward this postulated positive group, and charge neutralization takes place normally. In the Rp-isomer, however, the -P=S function points toward the positive group, and no charge neutralization and hence no reaction can take place. A second explanation involves steric hindrance for the Rp-isomer because of the presence of the bulky sulfur atom in this analog. P=S bond length is 1.9 to 2.0 Å, and P=O bond length is 1.6 to 1.7 Å.

Enzymatic addition involves inversion of the phosphorus configuration, thereby generating an internucleotide phosphate having the Rp configuration [54]. Thus, diastereomerically pure P(XY)-DNA and P(XY)-RNA can be obtained via the enzymatic route. This methodology has been applied to the synthesis of Rp phosphorothioate transcripts [55] and oligodeoxynucleotides with the help of modified T7 DNA polymerase [56] or *Taq* DNA polymerase [57]. Furthermore, the substitution of the oxygen atom with sulfur in 5′-(α-P-thio)triphosphates has been found to have a pronounced effect on their resistance to enzymes. The authors of work [19] cite as an example 5′-(α-P-thio)triphosphates of thymidine and uridine, which show resistance to calf intestinal alkaline phosphatase, as well as internucleotide-thiophosphate-containing thymidine dinucleotide, which is resistant to snake venom and spleen phosphodiesterases. Given that modified oligonucleotides can be obtained through the incorporation of α-thio-triphosphates into a nucleotide strand by polymerases, those authors [19] on the basis of their research hypothesized that an oligonucleotide completely modified with thiophosphate groups can also be resistant to some nucleases. Kunkel et al. [13] noticed that the incorporation of deoxycytidine thiotriphosphate (dCTPαS) into DNA enhances resistance to cleavage of the phosphorothioate internucleotide bond by the 3′-exonuclease activity of DNA polymerase I.

In work [58], it was confirmed that as compared to their natural counterparts, 5′-(α-P-thio)triphosphates of nucleosides are cleaved by nucleases more slowly. After detailed research on this phenomenon, differences in the sensitivity of individual diastereomers (Rp and Sp) to various nucleases were identified. For instance, ribonuclease T1 and snake venom phosphodiesterase cleave the Rp-diastereomer much more readily than they do the Sp one, which can be degraded mostly by the P1 nuclease (Figure 26) [59,60,61].

Thus, enzymatic hydrolysis can be utilized to identify phosphorothioate derivatives of nucleoside triphosphates of unknown diastereomeric composition.

Aside from their widespread use for the synthesis of phosphorothioate oligonucleotides, nucleoside 5′-(α-P-thio)triphosphates have been employed as tools for investigation into the mechanism of action of certain enzymes, for example, myosin and avian myeloblastosis virus reverse transcriptase [9]. Because phosphorothioate-containing polymers are resistant to the degradation by nucleases, and the sulfur atom imparts many favorable chemical properties, several applications in molecular biology have been devised that involve nucleoside 5′-(α-P-thio)triphosphates, including new methods of site-directed mutagenesis and DNA sequencing [62,63].

Thiophosphate modifications are also used in most aptamers modified on the ribose phosphate backbone. They are obtained by replacement of one or more native triphosphates with thiophosphates during SELEX. These thio derivatives of nucleotides are quite good substrates for DNA and RNA polymerases and effectively facilitate the creation of modified oligonucleotide libraries [7]. Simple substitution of a nonbridging oxygen atom with sulfur has important consequences because thio-aptamers, as already mentioned, are noticeably more resistant to nuclease degradation than unmodified aptamers and possess stronger affinity for protein targets because of a decrease in the magnitude of binding to cations and because of additional hydrophobic interactions with the protein [64,65,66]. On the other hand, the number of oxygen-to-sulfur substitutions in the aptamer backbone must be carefully thought through because the inclusion of too many thiophosphate groups may cause nonspecific binding and destabilize secondary structures of aptamers. Despite synthetic availability of nucleoside triphosphates containing two thiophosphate moieties, this modification became popular only after their use in SELEX [26,67,68].

### 3.2. Nucleoside 5′-(α-P-Borano)triphosphates

Nucleoside 5′-(α-P-borano)triphosphates (NTP*α*Bs) [16], just as thio-triphosphates, can easily compete with natural NTPs and can be incorporated into nucleic acids by means of DNA or RNA polymerases [13]. Moreover, for viral reverse transcriptases, some ddNTP analogs (e.g., dideoxy-NTP*α*B) have turned out to be better substrates (than conventional ddNTPs are) and may be promising prototypes of antiviral drugs as inhibitors of viral reverse transcriptases [69,70,71].

Oligonucleotides obtained enzymatically and carrying one or more boranophosphate modifications form stable duplexes with DNA [5,72] and can serve as templates in PCR [72].

The borane phosphate modification imparts unique characteristics to nucleotides and nucleic acids. When a borane group is introduced, a negative charge remains on the phosphate. The borane group is metabolically stable, and in this context, modified phosphates become more hydrophobic than native phosphate groups. Boranophosphate nucleotides and nucleic acids interact with proteins differently and penetrate cell membranes more efficiently than conventional phosphate or thiophosphate nucleotides can. When 5′-(α-P-borano)triphosphates are incorporated into DNA or RNA, the obtained oligonucleotides feature enhanced resistance to cellular nucleases [73]. Owing to these beneficial features, BH_3_-containing RNA aptamers to ATP have been obtained by direct SELEX-based selection using individual (α-P-borano)-modified triphosphates (UTPαB or GTPαB) [74].

The resistance of BH_3_ oligodeoxynucleotides to nucleolytic hydrolysis is a feature important for PCR during DNA sequencing [73]. A mixture of two stereoisomers of fully modified oligothymidylates has a lower melting point as compared to native ones; in this context, it is known that when bound to RNA, boranophosphate-modified nucleic acid analogs (just as thiophosphate-modified nucleic acid analogs) tend to increase the activity of RNase H; this effect makes them potentially useful as antisense oligonucleotides [73]. Boranophosphate-modified nucleotides hold promise for selective transport of boron into a tumor tissue for radiation therapy. If boron is localized to target cells, then these cells can be easily destroyed by boron neutron capture therapy [75].

### 3.3. Nucleoside 5′-(α-P-Alkyl)triphosphates

In contrast to boranophosphates, which retain negatively charged phosphodiester bonds of DNA, introduction of an alkyl (methyl or ethyl) group at the phosphorus atom causes the formation of a neutral phosphate group. It is reported that thymidine 5′-methylphosphonyl(diphosphate) (dTTPαCH_3_) is used by terminal deoxynucleotidyl transferase (TdT) to extend a 20-mer oligodeoxynucleotide primer by either one or two nucleotides at the 3′ end [40]. Subsequent degradation of the product using P1 nuclease and alkaline phosphatase followed by HPLC analysis revealed the presence of only two fragments: d[Cp(CH_3_)T] and d[Cp(CH_3_)Tp(CH_3_)T] (single-nucleotide and double-nucleotide extensions, respectively). This outcome indicates the resistance of methylphosphonate bonds to nucleases. These bonds are also resistant to the 3′-exonuclease activity of snake venom phosphodiesterase. The arrangement of the methylphosphonate bonds in the di- and trinucleotide fragments was determined beforehand to be the Sp configuration.

Thymidine 5′-(α-P-methyl)triphosphate has been tested as an alternative substrate for *Escherichia coli* DNA polymerase 1 (Klenow fragment) by means of several template systems requiring the formation of 1 to 42 methylphosphonodiester bonds. The enzyme catalyzes the insertion of a P-methyl-thymidyl residue having the (Sp) configuration [39].

A methylphosphonate bond in an oligonucleotide increases resistance to nuclease-mediated hydrolysis [76]. Studies suggest that a DNA–DNA duplex of a heptanucleotide with a heteronucleotide sequence composed of stereopure Rp-isomers of methylphosphonate oligonucleotides is stabler than the duplex containing at least one Sp-isomer [77]. Replacement of more Rp-isomers with Sp ones leads to further destabilization of the binding to a target DNA/RNA strand. Therefore, for use as effective antisense and antigen therapeutics, it is preferable to obtain chirally pure Rp-stereoisomers.

In the last few years, by synthetic chemistry and polymerase engineering, Holliger’s group [43] described enzymatic forward and reverse synthesis of trialkylphosphonate nucleic acids using a DNA template and a mixture of α-P-methyl- and α-P-ethyl-triphosphates as well as directed evolution of specific streptavidin-binding aptamers of trialkylphosphonate nucleic acids directly from a repertoire of mixed P-methyl/P-ethyl trialkylphosphonate nucleic acids of random sequence. Those authors noticed that a large number of ethylphosphonate groups in the oligomer reduces solubility in water and causes a loss of affinity for the complementary strand.

Overall, the above-mentioned disadvantages made alkyl modifications less attractive and their utility for therapeutic oligonucleotides less pronounced.

### 3.4. Nucleoside 5′-(α-P-Imido)triphosphates

Lately, the ability of a new triphosphate with the (α-P-imido) modification—namely thymidine 5′-(1,3-dimethylimidazolidin-2-ylidene)triphosphate (TTPαDMI)—to serve as a substrate in elongation reactions has been studied for two types of DNA polymerases: TdT and DNA polymerase I from *E. coli* (Klenow fragment, Pol l) [49]. Toward DNA polymerase I, imido-triphosphate TTPαDMI showed low substrate activity. At present, a method for template-independent enzymatic synthesis of oligonucleotides is being actively developed using the TdT enzyme [78]. Reversible triphosphate terminators are essential components in this approach to using enzymes for creating artificial single-stranded DNA. It has been demonstrated that TTPαDMI can serve as a substrate for human TdT, which is responsible for random addition of nucleotides to the 3′ end of a single-stranded DNA primer. Those authors found that TdT better recognizes native thymidine triphosphates, but without deoxythymidine triphosphate (dTTP) and with increasing concentration of a modified nucleoside triphosphate, recognition and incorporation of a single nucleotide into the DNA strand occur too. No further strand elongation took place, i.e., the modification almost completely blocked subsequent incorporation of even native nucleoside triphosphates [49]. Considering that the imido modification can be removed (chemically or enzymatically) after incorporation of the nucleotide into the strand, it is possible to design reversible terminators for TdT that are based on 5′-(α-P-imido)-modified nucleoside triphosphates.

Currently, substantial attention is given to topical research on effects of the stereospecificity of various phosphorus-modified blocks of oligonucleotides for their use in antisense technologies. For instance, thio- and phosphorylguanidine modifications have been examined: various isomers at different positions of both sense and antisense oligonucleotide strands. Judicious use of stereopure phosphorylguanidine linkages has improved the profile of mRNA silencing in mouse hepatocytes in vivo without disturbing endogenous RNA interference pathways and without increasing serum levels of biomarkers of liver dysfunction, suggesting that such oligos may be suitable for therapeutic use [79]. The obtaining of stereopure modifications in oligonucleotides has become possible due to the creation of new amidophosphites that ensure subsequent stereopure modification by means of an azide during oligonucleotide synthesis [80]. Obviously, the production of stereopure 5′-(α-P-imido)-modified triphosphates is a serious methodology that can take the place of enzymatic synthesis of such oligonucleotides.

## 4. Prospects of α-Phosphate-Modified Nucleotides for the Design of Mutant Enzymes

In 2018, three scientists shared the Nobel Prize in Chemistry: Frances H. Arnold “for directed evolution of enzymes” and George P. Smith and Sir Gregory P. Winter “for phage display of peptides and antibodies.” All these researchers have links to the development of techniques for producing useful proteins and peptides that are based on imitation of the natural “method” of biological evolution, namely, a combination of random variation and nonrandom selection, which determine progress in enzymatic synthesis and evolution of unnatural nucleic acids. The development of modern methods for construction of engineering polymerases enables investigators to obtain and study a huge set of new biopolymers as well as various ligands, catalysts, and materials based on these biopolymers. In most cases, artificial polymerases are used for enzymatic synthesis of nucleic acid analogs that do not occur in nature, thereby considerably expanding the possibilities related to modified nucleoside triphosphates.

For instance, triphosphates containing 5′-(α-P-methyl) and 5′-(α-P-ethyl) modifications have served as a tool for relevant research. It is generally accepted that physicochemical properties of nucleic acids depend on the negatively charged phosphodiester backbone. The polyelectrolyte structure has been proposed as a defining feature of all informational biopolymers (mRNA and DNA). Nonetheless, this hypothesis has not been tested experimentally. In work [43], investigators describe an example of DNA-templated enzymatic synthesis and evolution of uncharged genetic polymers and offer a fundamental methodology for studying them as a source of new functional molecules. In this genetic polymer, the canonical negatively charged internucleoside phosphodiester is replaced by an uncharged P-alkyl phosphonodiester backbone.

Using engineered polymerases, one can obtain artificial analogs of nucleic acids consisting of several hundred nucleotides: impossible with chemical synthesis [81]. The oligonucleotides synthesized in this way can be utilized as aptamers resistant to degradation in vivo [82], for the creation of ribozymes [83] and three-dimensional nanostructures [84], in xenobiological experiments on the creation of semisynthetic living organisms [85], and in other promising scientific fields.

## 5. Conclusions

Basic methods for the synthesis of nucleoside triphosphate derivatives modified at 5-α-phosphate were developed 30–40 years ago, when approaches to the synthesis of nucleoside triphosphates were rapidly developing. In the last decade, efforts in this area have somewhat weakened, but work on optimizing the reactions and increasing the product yields continues. As shown above, the synthetic approach via cyclic P^V^P^V^P^III^-nucleoside intermediates proposed by Ludwig and Eckstein, as the most convenient, could be further developed into an industrial-scale process.

Recent renewed interest in the production of these important compounds is due to their widespread use in various fields of both basic research and medical applications, e.g., for the synthesis of aptamers or antisense oligonucleotides as well as for sequencing and PCR. Widespread use of PCR diagnostics at present and advancements in modern methods for constructing artificial polymerases have uncovered the need for the design of new (and optimization of existing) techniques for the synthesis of α-phosphate-modified triphosphates. Especially promising are 5′-(α-P-thio)-, 5′-(α-P-borano)-, and 5′-(α-P-imido)-modified nucleoside triphosphates. The first two have already found broad applications in antisense oligonucleotide therapies, sequencing, and PCR. There are efficient protocols for their synthesis. On the other hand, imido-triphosphate derivatives were proposed quite recently, and routes of their synthesis are not yet optimal. Nonetheless, the diversity of modifications that can be implemented through Staudinger’s reaction with various organic azides appears to be large. Such nucleotide analogs will certainly be helpful for basic research on mechanisms of enzyme action, including the formation of reversible and irreversible complexes with enzymes and enzyme inhibition. These analogs can also be effective in PCR diagnostics and in the synthesis of therapeutic oligonucleotides, for the purpose of increasing specificity, changing the interaction of oligonucleotides with complementary strands of nucleic acids, improving the efficiency of intracellular delivery, and for other applications. First experiments showed potential usefulness of imidophosphate derivatives for template-independent enzymatic synthesis of oligonucleotides by means of TdT.

This review comprehensively categorized data on chemical methods of synthesis of α-phosphate-modified nucleoside triphosphates (α-phosphate mimetics) and on their substrate properties toward nucleic acid metabolism enzymes. The main areas of their applications were analyzed as were the uses of oligonucleotides that can be generated from modified (d)NTPs. The review will be helpful to synthetic chemists designing new nucleic acid derivatives, to molecular biologists studying enzymatic processes, and to biomedical researchers employing modified nucleoside triphosphates and oligonucleotides for molecular–diagnostic solutions and for the development of new therapeutic compounds.

## Data Availability

Not applicable.

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
