# Peer review of "Synthesis and Properties of α-Phosphate-Modified Nucleoside Triphosphates"

_molecules, 2024, doi:10.3390/molecules29174121_

Round 1

Reviewer 1 Report

Comments and Suggestions for Authors

This review focuses on chemical methods for the synthesis of α‐phosphate–modified nucleoside triphosphates (α‐phosphate mimetics) and on their substrate properties toward nucleic‐acid metabolism enzymes. Prospects of α‐phosphate–modified nucleotides for the development of mutant enzymes are also reported.

The manuscript can be accepted for publication, after English editing and following the comments below:

Figure 1 The authors must indicate what substituent R1 and R2 correspond to.

The authors should add the numbers of all the compounds in the description of the figures for example:

Line 161: add the number of the nucleoside (7)

Line 183: insert compound number (14) after and in line 185 (compounds 15,16)

Line 192: insert compound no (18) after cyclophosphate, and in line 195 (compound 20)

Line 291: give final compound no

Figures 16, 18: use similar drawing settings in all Figures

Comments on the Quality of English Language

Moderate English editing required

Author Response

We thank Reviewer 1 for your valuable comments. We have attempted to correct any issues.

Comments 1. Figure 1 The authors must indicate what substituent R1 and R2 correspond to.

Response 1: We corrected Scheme 1 (in previous version Figure 1).

Comments 2. The authors should add the numbers of all the compounds in the description of the figures for example:

Line 161: add the number of the nucleoside (7)

Line 183: insert compound number (14) after and in line 185 (compounds 15,16)

Line 192: insert compound no (18) after cyclophosphate, and in line 195 (compound 20)

Response 2: Done. Changes in the text are highlighted in yellow.

Reviewer 2 Report

Comments and Suggestions for Authors

In this manuscript, “Synthesis and Properties of α-Phosphate-Modified Nucleoside 2

Triphosphates” by Novgorodtseva et al. summarizes information about chemical methods for the synthesis of nucleoside triphosphates modified on α‐phosphate (α‐phosphate mimetics) and their substrate properties toward nucleic‐acid metabolism enzymes. Although authors had collected all literature, this work seems premature for publication. Therefore, I would suggest that authors may take a major revision. Here are the comments and suggestions:

1.      The applications of α-Phosphate-Modified Nucleoside Triphosphates from literature  should be added.

2.      Some Tables can be also added for the comparison of their substrate properties and/or their applications.

Author Response

We would like to thank Reviewer 2 for the attention to our work and useful suggestions. We have tried to bring clarity to our work and take into account all your recommendations.

Comments 1. The applications of α-Phosphate-Modified Nucleoside Triphosphates from literature should be added.

Response 1: Chapter 3 « Substrate properties of 5′-α-P-modified nucleoside triphosphates and of oligonucleotides synthesized from them in relation to nucleic-acid processing enzymes. The applications of α-phosphate-modified nucleoside triphosphates» (pages 21-27) discusses the properties and applications of α-Phosphate-Modified Nucleoside Triphosphates.

Comments 2. Some Tables can be also added for the comparison of their substrate properties and/or their applications.

Response 2: In the review, substrate properties of various α-P-modified nucleoside triphosphates are described. In analyzed researches, a lot of enzymes interactions with various NTPs have been studied. Moreover, these works have been carried out under various conditions (buffer, concentrations, etc.). All together, these significantly complicate a systematic and quantitative analysis of the substrate properties of the modified NTPs in this paper. Such a study can be an object of distinct review.

Reviewer 3 Report

Comments and Suggestions for Authors

Synthesis and Properties of α-Phosphate-Modified Nucleoside

Triphosphates

The Authors  Alexander A. Lomzov and Svetlana V. Vasilyeva are reporting a review about chemical methods for the synthesis of nucleoside triphosphates modified on α‐phosphate (oxygen atoms of α‐phosphate was replaced with sulfur, selenium, borane, alkyl, amine, N‐alkyl, or another substituent) and their usefulness in studying functions of the enzymes, investigating the stereochemical course, and mechanisms of the enzymatic reactions.

Comments:

1.     Page 1, lines 30-33, author should provide citation.

2.     Page 1, lines 33-35, author should provide citation.

3.     Page 1, lines 35-38, author should rephrase the sentence.

4.     In introduction authors should discuss about difficulty in introducing heteroatoms at α-P position NTPs. Also, should discuss about two major types of synthetic approaches; 1. Coupling of pyrophosphate with activated forms of modified NMP and 2. Use  of cyclic PVPVPIII-nucleosides are intermediates for the synthesis of NTPαX and include a scheme showcasing the two methods.

5.     Page 2, lines 71-74, author can explain in more detail regarding why DNA and RNA polymerases “recognize” only certain diastereomers of nucleotide analogs?

6.     Page 3, lines 110-113, for reference 16, author should include a representative scheme and rephrase the sentences.

7.     Page 4, lines 121-125, author should provide citation.

8.     For Auer. M. et. al. work author should include a representative scheme.

9.     Page 4, lines 128, provide the name of the corresponding author from the journal.

10.  For reference 20, author should include a representative scheme.

11.  Page 4, lines 137, remove the word “so-called”.

12.  Page 4, lines 141-148, rephrase the sentences.

13.  Figure 4, labelling is confusing. B stands for different bases (A, G, C, T or U), what  Bi stand for and what is BBi ? Also, why there is a R1 in structure 6?

14.  Page 5, lines 165, remove “Those authors confirmed that the ring opening occurs” to  “The authors confirmed that the ring opening occurs”.

15.  Page 5, lines 175-178, rephrase the sentence.

16.  Page 5, lines 179-182, rephrase the sentence.

17.  Figure 5, labelling is confusing. Does 15 corresponding to  NTPαS and 16 corresponding to  NTPαSe?

18.  Page 7, lines 206-208, rephrase the sentence.

19.  For chapter 2.1 the author should include the work of Qi Sun (Asian J. Org. Chem. 2020, 9, e202200696).

20.  Page 7, lines 211, “via β‐elimination of the cyanoethyl group as the mechanism” to “via β‐elimination of the cyanoethyl group”.

21.  Page 7, line 223, “giving a mixture of diastereomers of TTPαB 30” to “giving a mixture of diastereomers of TTPαB and TTβB which was purified ….

22.  Page 7, line 225-227, is confusing, rephrase the sentence.

23.  Page 8, line 232-233, rephrase the sentence.

24.   Page 8, line 232-233, “adenosine analogs modified in the heterocycle” to ”adenosine analogs with modification in the heterocycle ring”.

25.  Figure 10, redraw the triphosphate structure.

26.  Page 8, line 247-250, rephrase the sentence.

27.  Figure 11, redraw the triphosphate structure.

28.  Page 9, line 258, “Me2S:BH3 was used as a modifying reagent” to “Me2S:BH3 was used as a borating reagent”.

29.  Figure 12, what was diastereomeric ratio of 36 obtained?

30.  For reference 34, author should include a representative scheme.

31.  Author should include the synthesis of thymidine 5‐methylphosphonate in figure 13.

32.  Author should reorganize the scheme in figure 15.

33.  Author should author should include a representative scheme for reference 42 and provide patent number for reference 42.

34.  Page 12, line 355-357, rephrase the sentence and ‘R’ should be electron withdrawing group.

35.  Page 12, line 364-366, author was discussing synthesis of adenosine 5‐(α‐P‐imido)triphosphate but figure 17 represents thymidine.  

36.  Page 13, line 386-388, rephrase the sentence.

37.  Figure 16 and 19 structures should be resized to other structures.

38.  Instead of figure rename them to scheme which will be more appropriate.

Comments on the Quality of English Language

 Minor editing of English language required.

Author Response

RE: Revised manuscript molecules-3137596

Response to Reviewer 3

We thank Reviewer 3 for careful reading of our manuscript. Thank you for your comments and suggestions, we tried to take them into account fully. We hope that this significantly improved our review.

Reviewer 3:

Comment 1-3:

  1. Page 1, lines 30-33, author should provide citation.
  2. Page 1, lines 33-35, author should provide citation.
  3. Page 1, lines 35-38, author should rephrase the sentence.

Response: Done. We have significantly changed the Introduction and provided citations. Changes in the text are highlighted in yellow.

Comment 4. In introduction authors should discuss about difficulty in introducing heteroatoms at α-P position NTPs. Also, should discuss about two major types of synthetic approaches; 1. Coupling of pyrophosphate with activated forms of modified NMP and 2. Use of cyclic PVPVPIII-nucleosides are intermediates for the synthesis of NTPαX and include a scheme showcasing the two methods.

Response: Our review is focuses on the progress achieved in the synthesis of α-P-modified nucleoside triphosphates (α-phosphate mimetics). However, the paper is also highlights their substrate properties for nucleic acid metabolism enzymes and some of the most important applications for them. Therefore, we have moved the discussion about two major types of synthetic approaches to the end of Part 2."Chemical synthesis of nucleoside 5'-α-P–modified triphosphates" under the heading ″2.6. Summary″.

Comment 5. Page 2, lines 71-74, author can explain in more detail regarding why DNA and RNA polymerases “recognize” only certain diastereomers of nucleotide analogs?

Response: We added explanation of this fact in part 3.″Substrate properties of 5'-α-P-modified ...″. Page 22, lines 503-518.

Comment 6. Page 3, lines 110-113, for reference 16, author should include a representative scheme and rephrase the sentences.

Response: Done. Scheme 3.

Comment 7. Page 4, lines 121-125, author should provide citation.

Response: Done.

Comment 8. For Auer. M. et. al. work author should include a representative scheme.

Response: Auer. M. et. al. refer in their work to work [21] which is represented in Scheme 4.

An additional scheme was not needed.

Comment 9. Page 4, lines 128, provide the name of the corresponding author from the journal.

Response: Done.

Comment 10. For reference 20, author should include a representative scheme.

Response: Done. Scheme 5.

Comment 11. Page 4, lines 137, remove the word “so-called”.

Response: Done.

Comment 12. Page 4, lines 141-148, rephrase the sentences.

Response: Done.

Comment 13. Figure 4, labelling is confusing. B stands for different bases (A, G, C, T or U), what Bi stand for and what is BBi ? Also, why there is a R1 in structure 6?

Response: Corrected.

Comment 14. Page 5, lines 165, remove “Those authors confirmed that the ring opening occurs” to “The authors confirmed that the ring opening occurs”.

Response: Done.

Comment 15. Page 5, lines 175-178, rephrase the sentence.

Response: Done.

Comment 16. Page 5, lines 179-182, rephrase the sentence.

Response: Done.

Comment 17. Figure 5, labelling is confusing. Does 15 corresponding to NTPαS and 16 corresponding to NTPαSe?

Response: Corrected.

Comment 18. Page 7, lines 206-208, rephrase the sentence.

Response: Done.

Comment 19. For chapter 2.1 the author should include the work of Qi Sun (Asian J. Org. Chem. 2020, 9, e202200696).

Response: Done.

Comment 20. Page 7, lines 211, “via β‐elimination of the cyanoethyl group as the mechanism” to “via β‐elimination of the cyanoethyl group”.

Response: Done.

Comment 21. Page 7, line 223, “giving a mixture of diastereomers of TTPαB 30” to “giving a mixture of diastereomers of TTPαB and TTβB which was purified ….

Response: In this case, the only information on α phosphate (first phosphorus atom in triphosphate, see Scheme 1) is presented. The diastereomers in the review are denoted as Sp and Rp (see in Part 3).

Comment 22. Page 7, line 225-227, is confusing, rephrase the sentence.

Response: Done.

Comment 23. Page 8, line 232-233, rephrase the sentence.

Response: Done.

Comment 24. Page 8, line 232-233, “adenosine analogs modified in the heterocycle” to ”adenosine analogs with modification in the heterocycle ring”.

Response: Corrected.

Comment 25. Figure 10, redraw the triphosphate structure.

Response: Done. Figure 10→ Scheme 13.

Comment 26. Page 8, line 247-250, rephrase the sentence.

Response: Done.

Comment 27. Figure 11, redraw the triphosphate structure.

Response: Done. Figure 11→ Scheme 14.

Comment 28. Page 9, line 258, “Me2S:BH3 was used as a modifying reagent” to “Me2S:BH3 was used as a borating reagent”.

Response: Corrected.

Comment 29. Figure 12, what was diastereomeric ratio of 36 obtained?

Response: Done. An diastereomeric ratio has been added in the text, Page 12, lines 267-272

Comment 30. For reference 34, author should include a representative scheme.

Response: Done. Scheme 16.

Comment 31. Author should include the synthesis of thymidine 5‐methylphosphonate in figure 13.

Response: Figure 13→ Scheme 17. In work [39] (earlier 35) the scheme of thymidine 5-methylphosphonate synthesis is not presented, and there is no description of it. Thymidine 5-methylphosphonate can be obtained as shown in Scheme 18, for example.

Comment 32. Author should reorganize the scheme in figure 15.

Response: Done.

Comment 33. Author should author should include a representative scheme for reference 42 and provide patent number for reference 42.

Response: Done.

Comment 34. Page 12, line 355-357, rephrase the sentence and ‘R’ should be electron withdrawing group.

Response: Done.

Comment 35. Page 12, line 364-366, author was discussing synthesis of adenosine 5‐(α‐P‐imido)triphosphate but figure 17 represents thymidine.

Response: Corrected.

Comment 36. Page 13, line 386-388, rephrase the sentence.

Response: Done.

Comment 37. Figure 16 and 19 structures should be resized to other structures.

Response: Done.

Comment 38. Instead of figure rename them to scheme which will be more appropriate.

Response: Done.

Reviewer 4 Report

Comments and Suggestions for Authors

This is a review article.

It is well written but a table of contents at the begining would be a good adition.

I would also recommend that the schemes be drawn in CD and not taken from the cited literature, as this diminishes their quality. If they are drawn the quality and resolution needs to be better. All structures need to be of the same size, having different sizes is not appropriate and leaves a bad impretion.

In the schemes of the synthesis that are in the paper it would be benefitial to add YIELDS, at least at the end  of the overall synthetic path. As this article is about different routes this is important to show. It would also be advasible to have a comparative table with yields for the same product (or functionality) via different routes.

In the conclusion part I'm missing a conclusion as to what methods give higher yields and/or products that are more pure. Which could be further developed into process scales.

As the review is aimed at synthetic chemists the remarks that I have given are important to implement, especially the yields and some kind of visual and/or table comparisons on the yields and the scales that these synthessis have been done in.

Comments on the Quality of English Language

fine

Author Response

RE: Revised manuscript molecules-3137596

Response to Reviewer 4

We thank Reviewer 4 for your comments and suggestions, we take them into account.

Reviewer 4:

Comments 1. ″ ...a table of contents at the begining would be a good adition.″

Response: We agree with the reviewer, however, according to the rules for authors, when submitting a review article to the journal, a table of contents is not provided.

Comments 2.″I would also recommend that the schemes be drawn in CD and not taken from the cited literature, as this diminishes their quality. If they are drawn the quality and resolution needs to be better. All structures need to be of the same size, having different sizes is not appropriate and leaves a bad impretion. ″

Response: Done. All figures in the manuscript have the same size and a good quality.

Comments 3.″In the schemes of the synthesis that are in the paper it would be benefitial to add YIELDS, at least at the end of the overall synthetic path. As this article is about different routes this is important to show. It would also be advasible to have a comparative table with yields for the same product (or functionality) via different routes. ″

Response: Done. We have added the yields of the described compounds to the text. We also added “Table 1. The synthetic approaches for the preparation of α-P-modified nucleoside triphos-phates (NTPαXYs)″ in the text. This is convenient for comparing synthesis methods and product yields.

Comments 4.″In the conclusion part I'm missing a conclusion as to what methods give higher yields and/or products that are more pure. Which could be further developed into process scales.″

Response: We have added the discussion about major types of synthetic approaches to the end of Part 2."Chemical synthesis of nucleoside 5'-α-P–modified triphosphates" in the section ″2.6. Summary″.

Comments 5.″As the review is aimed at synthetic chemists the remarks that I have given are important to implement, especially the yields and some kind of visual and/or table comparisons on the yields and the scales that these synthessis have been done in.″

Response: Done. Information on the yields is added in the text and summarized in Table 1.

Round 2

Reviewer 2 Report

Comments and Suggestions for Authors

It seems more acceptable now.

Reviewer 3 Report

Comments and Suggestions for Authors

Synthesis and Properties of α-Phosphate-Modified Nucleoside

Triphosphates

The Authors  Alexander A. Lomzov and Svetlana V. Vasilyeva are reporting a review about chemical methods for the synthesis of nucleoside triphosphates modified on α‐phosphate (oxygen atoms of α‐phosphate was replaced with sulfur, selenium, borane, alkyl, amine, N‐alkyl, or another substituent) and their usefulness in studying functions of the enzymes, investigating the stereochemical course, and mechanisms of the enzymatic reactions.

Comments:

1.     The authors responded and revised the proposed point by point as required.

Overall Comment: Accept.